# A New Clinical Protocol for a Timely Diagnosis and Treatment of Hydrocephalus in Patients with Severe Acquired Brain Injury

**DOI:** 10.3390/brainsci13071067

**Published:** 2023-07-13

**Authors:** Francesca Cesira Cava, Giovanna Barbara Castellani, Elisa Maietti, Pamela Salucci, Valentina Colombo, Giorgio Palandri

**Affiliations:** 1Montecatone Rehabilitation Institute, 40026 Imola, Italy; francesca.cava@montecatone.com (F.C.C.); pamela.salucci@montecatone.com (P.S.); valentina.colombo@montecatone.com (V.C.); 2Department of Biomedical and Neuromotor Sciences, Alma Mater Studiorum University of Bologna, 40126 Bologna, Italy; elisa.maietti2@unibo.it; 3Department of Neurosurgery, Institute of Neurological Sciences of Bologna IRCCS, Bellaria Hospital, 40139 Bologna, Italy; giopalandri@gmail.com

**Keywords:** rehabilitation outcome, brain injuries, hydrocephalus

## Abstract

Background: Secondary hydrocephalus is a well-known complication of severe acquired brain injuries (sABIs) often diagnosed during inpatient rehabilitation. Currently, there is no gold standard for its detection. Therefore, we designed a novel clinical diagnostic protocol that integrates clinical, functional, biochemical and neuroradiological assessments to improve the accuracy of its diagnosis in patients with sABIs. Methods: This prospective cohort study will be conducted in a tertiary referral rehabilitation center in Italy. A historical cohort of patients will be compared with a prospective cohort undergoing the new clinical diagnostic protocol. Expected Results: The expected results include an increase in the proportion of diagnosed cases, a reduced incidence of clinical complications, an increase in the rehabilitative outcomes at discharge, a significant reduction in the length of hospital stay, and useful information about the diagnostic and prognostic value of the neuroradiological characteristics. Conclusion: We expect that this clinical diagnostic protocol will result in a more appropriate assessment and timely treatment of secondary hydrocephalus in patients with sABIs, with the ultimate goal of improving their prognosis. In addition, it could be adopted by other rehabilitation centers to improve hydrocephalus diagnosis and treatment, thereby reducing the length of hospital stay and accelerating recovery with benefits for both patients and hospitals.

## 1. Introduction

Secondary hydrocephalus is a pathological condition caused by the accumulation of cerebrospinal fluid (CSF), probably as a consequence of damage in its reabsorption mechanisms. It is a well-known complication of severe acquired brain injuries (sABIs), often diagnosed during inpatient rehabilitation [1,2,3]. The incidence of chronic secondary hydrocephalus is reported to range between 14 and 45%, and it is usually diagnosed between 3 and 12 months after brain injury [1,4,5,6,7].

Detecting secondary hydrocephalus in patients with sABIs is complex, due to their neurological impairment, difficulties in autonomously reporting symptoms and the actual lack of standardized methods of diagnosis. Moreover, even when a clear ventriculomegaly is found on CT or RMN scans, it is sometimes difficult to understand whether it is secondary to brain atrophy or the accumulation of CSF [1,2,5]. Hydrocephalus in patients with sABIs can present either with symptoms related to chronic hydrocephalus (cognitive impairment, gait disturbances and urinary incontinence) or with features more related to increased intracranial pressure (headache, papilledema, nausea, vomit and worsening of consciousness) [5,8]. In the absence of a gold standard for the diagnosis of secondary hydrocephalus, its recognition is often based on the combination of clinical observation and radiological findings. Subarachnoid and/or intraventricular hemorrhage, decompressive craniectomy, duration of coma, severity of brain injury (measured using the Glasgow Coma Scale score) and older age are reported as the main risk factors for hydrocephalus [4,6,9,10]. The main diagnostic criteria include a persistent disorder of consciousness, worsened or plateaued clinical recovery, elevated CSF pressure and/or clinical improvement in behavioral or motor capacities after high-volume lumbar puncture [3,11,12].

Given the difficulties in its diagnosis and treatment, the identification of hydrocephalus is important to clinicians dealing with the rehabilitation of patients with sABIs because it may delay or negatively affect their recovery. Moreover, being a treatable condition, early diagnosis and correct management are fundamental to identify patients who may be selected for surgical treatment, which is associated with a better outcome [2,4,6,10,11,13,14,15,16]. Surgical treatment usually consists in the placement of a ventriculoperitoneal shunt (VPS) or ventriculoatrial shunt (VAS) [13,14,15].

In a previous study, a 13.2% incidence of secondary hydrocephalus was reported among 621 hospitalized patients with sABIs [7]. The diagnosis was based on the previously indicated combination of clinical and radiological criteria, but we hypothesized that cases may have been under-detected, especially in patients with disorders of consciousness and/or many confounding comorbidities, due to the absence of a reference standard. In addition, the study showed that after VPS placement, most patients had a clinical improvement and none worsened; however, a delay in surgical treatment resulted in a longer hospitalization and was associated with worse outcomes. That study underscored the need for a more standardized approach to the diagnosis of secondary hydrocephalus, with a better timing in the evaluation of patients with sABIs during inpatient rehabilitation in order to improve the selection of those who may benefit most from surgical treatment [17].

In this position paper, we describe a diagnostic protocol for hydrocephalus that integrates clinical, functional, biochemical and neuroradiological assessments. The aims of the protocol are as follows:To increase the hydrocephalus detection rate;To reduce the time for hydrocephalus diagnosis/surgical treatment;To improve the clinical rehabilitative outcomes at discharge;To reduce the overall length of hospitalization (LOS) and the occurrence of complications in patients diagnosed with and treated for hydrocephalus.

## 2. Materials and Methods

### 2.1. Study Design, Setting and Population

The protocol will be applied to a prospective cohort recruited in a tertiary referral rehabilitation center in Italy, highly specialized in sABIs. Patients aged 18 or older with a diagnosis of an sABI of any etiology will be eligible for the study. Exclusion criteria are as follows: diagnosis and/or treatment of hydrocephalus prior to hospital admission at our institution; presence of acute hydrocephalus; pre-existing neurological or psychiatric comorbidities. 

Patients with and without a diagnosis of hydrocephalus in this prospective cohort will be compared in terms of clinical rehabilitative outcomes, the evolution of quality of life and neuroradiological characteristics.

A historical cohort will serve as a control group and will include all hospitalized patients with sABIs between 1 January 2015 and 31 December 2019. The years 2020 and 2021 will be excluded in order to avoid selection bias due to the organizational measures taken to counteract the impact of the SARS-CoV-2 pandemic. 

### 2.2. Feasibility 

Patients with sABIs of different etiologies (traumatic, vascular, anoxic, infectious, neoplastic, etc.) will be recruited from those hospitalized in the Severe Acquired Brain Injury Unit or in the Intensive Care Unit (ICU), depending on their clinical conditions. The presence of an ICU in our rehabilitation institute ensures prompt treatment soon after the acute brain injury event, through the implementation of a personalized rehabilitation project in the early stage.

The recruitment of the prospective cohort will last two years. The expected annual flow of patients with sABIs is 50–60, and the expected proportion with hydrocephalus is about 20%, leading to a total number of 100–120 patients with sABIs and at least 20 patients with a confirmed diagnosis of hydrocephalus.

### 2.3. Ethics 

The study protocol has been approved by the local ethics committee (CE-AVEC N° 191-2022-OSS-AUSLIM; 17 March 2022), and the study did not receive funding. All participants will sign an informed consent form to participate in the study. In case of an inability to give consent, this will be acquired from the caregiver according to article 4 of Law n. 219, 22 December 2017. In the historical cohort, informed consent will be obtained whenever possible; otherwise, it will be waived in accordance with the General Authorization of the Privacy Guarantor No. 09/2016 for observational retrospective studies.

### 2.4. Clinical Diagnostic Protocol and Study Assessments

A multidisciplinary team of physiatrists, neuroradiologists, neuropsychologists and neurosurgeons designed the new protocol for the diagnosis of suspected secondary hydrocephalus and its confirmation, based on the best recommendations in the literature [18,19,20]. The timing and procedures included in the diagnostic protocol are shown in Figure 1. 

Within 7 days (week 1) of hospital admission, i.e., at baseline (T0), patients with sABIs will undergo the following evaluations:Socio-demographic and anamnestic data collection;Neurological examination;Clinical and functional assessment;Neuroradiological exam (brain CT);Neurophysiological exams.

Clinical and functional assessments will be performed at weeks 3 and 5, before the multidisciplinary team meeting. An additional CT scan and a brain MRI will be performed after 30 days (week 5) from admission (T1) to assess the presence of ventricular dilation signs. The first multidisciplinary team meeting will be held about 40 days (week 6) after admission, with the aim of evaluating the results of the neurological examination, neuroradiological exams, and clinical and functional assessments. The multidisciplinary team will identify patients with suspected hydrocephalus based on the presence of at least one of the following criteria:Neuroradiological criteria: increase in ventricular dimensions with supratentorial and subtentorial ventricular dilation, or both; sulcal reduction at the convexity; Evan’s index increase; callosal thickness reduction; callosal angle increase; presence of periventricular white matter changes; presence of aqueductal flow void.Clinical criteria: worsening or stopping of clinical recovery as assessed based on clinical scales and functional tests.

Patients with “suspected hydrocephalus” will undergo further evaluations to confirm the diagnosis, specifically:High-volume tap test (TT);CSF examination;Clinical and functional assessments 1, 3 and 7 days after TT;Multidisciplinary team evaluation 7 days after TT.

Diagnosis confirmation derives mainly from the improvement of the patient’s clinical/neurological condition after CSF TT, measured as an improvement in the score of the clinical scales before and after the test or as an improvement in the performances in the functional tests. The main clinical rating scale is the Disability Rating Scale (DRS). The Levels of Cognitive Functioning (LCF) scale corroborates the DRS result. The Coma Recovery Scale Revised (CRS-R) will be used for patients with low responsiveness or severe disturbance of consciousness. 

Specifically, the diagnosis will be confirmed when at least one of the following changes occurs after TT:
Decrease in the DRS score;Increase in the LCF score;Increase in one of the CRS-R scale’s items that identify a minimally conscious state or emergence from a minimally conscious state;Improvement in one of the functional tests (TUG, 10 mWT, POMA).

Patients with a confirmed diagnosis will receive an indication of surgical treatment with shunt placement.

Both the severity of the clinical condition and the consent being denied by family are reasons that preclude the intervention, even in the case of a diagnosis confirmation.

During the hospital stay, the multidisciplinary team will evaluate the presence of suspected hydrocephalus in patients with no previous diagnosis. Patients with suspected hydrocephalus will be treated following the clinical diagnostic protocol just described.

The study flow is described in Figure 2. 

All patients will undergo a reassessment of all clinical and functional scales, neurological and neurophysiological examinations and a brain CT scan at three time points: 1 month after surgery for those treated with a VPS/VAS or at 3 months from hospital admission for patients without a diagnosis of hydrocephalus or who did not receive a VPS/VAS placement (T2); at discharge (T3); and 1 year after hospitalization (T4). All the patients will undergo a brain MRI at T3 and, if prescribed by the multidisciplinary team, at T4. 

Table 1 summarizes the clinical parameters, clinical measurement scales, functional tests, and neurophysiological and neuroradiological exams included in the study.

### 2.5. Statistical Analysis Plan

Continuous variables will be presented as the mean and standard deviation or as the median and interquartile range. Normality in the distribution will be tested with the Shapiro–Wilk test. Categorical variables will be summarized in terms of absolute and relative frequencies.

To determine whether the new diagnostic protocol can improve the ability to detect hydrocephalus as well as the timeliness of the diagnosis and the clinical rehabilitative outcomes, the historical and prospective cohorts will be compared with respect to the following: hydrocephalus diagnosis detection rate, time from admission to diagnosis, LOS, incidence of clinical complications, and DRS and LCF scores at discharge (T3). In the subgroup of patients with low responsiveness, the clinical rehabilitative outcome at discharge will be measured using the CRS-R score. Continuous variables will be compared using Student’s *t* test, when normally distributed, or the Mann–Whitney U test otherwise. Categorical variables will be compared using Pearson’s χ^2^ test or Fisher’s exact test as appropriate. The same tests will be used to compare patients with and patients without a diagnosis of hydrocephalus belonging to the prospective cohort, in terms of the following: clinical rehabilitative outcome at T3 and T4 measured using the DRS, LCF and CRS-R scale scores; evolution from T0 to T4 of quality of life measured using the QOLIBRI-OS; neuroradiological characteristics at T1 and T3 or T4 measured using Evan’s index, callosal angle and volumetric analysis. To complement the information provided by the statistical tests, differences between groups will be quantified in terms of the effect size (standardized mean differences for continuous variables; difference between proportions or odds ratios for categorical variables) and its standard error. In addition, outcome variables will be compared between the historical and prospective cohorts, adjusting for demographic and clinical confounders in multivariable regression analyses. Confounders to be included in the models are those significantly associated with the outcomes and significantly different between the two cohorts. 

Two-sided tests will be carried out, and the statistical significance level will be set at 0.05. All analyses will be performed with Stata version 17.0 (Stata Corp., College Station, TX, USA).

## 3. Expected Results

We expect to find, in the prospective cohort as compared to the historical cohort, an increase in the proportion of diagnosed cases, a significant reduction in the time elapsed between admission and diagnosis and, consequently, a shortened time to surgery as well as a shortened overall LOS.

We assume that early diagnosis and treatment may result in a decreased incidence of clinical complications and may improve the rehabilitative outcomes at discharge among patients with hydrocephalus, especially in the subgroup with low responsiveness.

In addition, we expect to obtain useful information about the diagnostic and prognostic value of the neuroradiological characteristics.

Lastly, from the comparison of patients with and without hydrocephalus in the prospective cohort, we expect to observe that the adoption of an objective and timely protocol for the diagnosis and treatment will allow patients with hydrocephalus to obtain similar clinical rehabilitative outcomes and perceived quality of life to those who do not develop this complication.

## 4. Discussion

Secondary hydrocephalus is a well-known complication of sABIs that may delay or negatively affect patients’ recovery, apparently resulting from multiple factors. Currently, there is no gold standard for its detection, but early diagnosis and treatment are fundamental to improve patients’ prognosis and rehabilitation outcomes [20,44]. In this paper, we have described the study protocol of a prospective cohort study that aims at validating a new clinical protocol for the diagnosis of secondary hydrocephalus. 

The liquor subtraction test is essential for the diagnosis [5,8,45]; however, because of its transient effect, a standardized and timely evaluation is mandatory.

Secondary hydrocephalus is under-diagnosed and less studied compared to idiopathic normal-pressure hydrocephalus (iNPH). In a recent retrospective study, Lalou et al. [44] found that resistance to CSF outflow and the pulse amplitude of intracranial pressure were significantly lower in PTH compared to iNPH and did not always reflect the degree of hydrocephalus or atrophy reported on CT/MRI. More studies on CSF dynamics are needed in order to assess the CSF circulation between people with trauma or hemorrhage compared with people with iNPH. 

Wu and colleagues [46] described secondary low-pressure hydrocephalus (LPH) and negative-pressure hydrocephalus (NegPH) that have been rarely reported and insufficiently recognized pathophysiologically. They suggested that, although the current classifications of hydrocephalus fail to provide clinical guidance to manage LPH and NegPH properly, they offer a new therapeutic viewpoint in order to recover the ventricular enlargement back to a normal size as far as possible from hydrocephalus with ventriculomegaly, regardless of the ventricular pressure. 

Given these premises, we plan to systematically submit all patients with sABIs to a timely and structured protocol that includes a clinical assessment, tap test, CSF examination, and neurophysiological and neuroradiological exams. In our protocol, neuroimaging is performed according to the timing based on the literature and our experience, in order to capture the hydrocephalus as soon as possible. 

Some patients with sABIs present with prolonged disorders of consciousness (unresponsive wakefulness syndrome or minimal conscious state). Arnts and colleagues [19] discussed both the difficulties in the diagnosis and the dilemmas in the treatment of CSF disorders in patients with prolonged disorders of consciousness and reviewed evidence from the literature to implement an active surveillance protocol for the detection of this late but treatable complication. Moreover, they advocated a low threshold for cerebrospinal fluid diversion when hydrocephalus is suspected, even months or years after brain injury. In addition, an accurate and structured diagnostic investigation with CT and MRI allows the identification of patients clinically not suspected of hydrocephalus. 

It is generally recognized that a multidisciplinary approach plays an important role in the management and treatment of suspected hydrocephalus cases, because risks and benefits should be carefully considered and calibrated on the patient [1,7]. In our protocol, the multidisciplinary team represents the most important feature since the exchange of views among experts allows for evaluating patients’ conditions in deeper detail and identifying the best care pathway.

This new clinical diagnostic protocol has been designed to integrate all the aforementioned ingredients. 

One important limitation is that the new protocol has already been adopted by our institution; thus, carrying out a randomized clinical trial to assess its efficacy is not feasible for ethical reasons. The second limitation is that the historical cohort may not include complete information on the variables of interest. 

## 5. Conclusions

We designed a novel clinical diagnostic protocol to improve the accuracy of secondary hydrocephalus diagnosis in patients with sABIs. We expect that it will result in a more appropriate assessment and timely treatment of patients, with the ultimate goal of improving their prognosis.

Should this clinical diagnostic protocol prove to be useful, it could be adopted by other rehabilitation centers to improve hydrocephalus diagnosis and treatment, thereby reducing the length of hospital stay and accelerating recovery with benefits for both patients and hospitals. 

In addition, this protocol may inform future diagnostic and prognostic guidelines for the inclusion of the tap test, timely and standardized neuroimaging and an early approach of the multidisciplinary team in the standard of secondary hydrocephalus diagnosis in patients with sABIs. 

## Figures and Tables

**Figure 1 brainsci-13-01067-f001:**
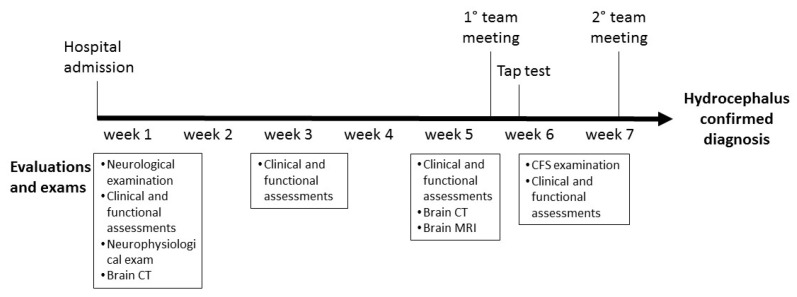
Diagnostic protocol timing and procedures.

**Figure 2 brainsci-13-01067-f002:**
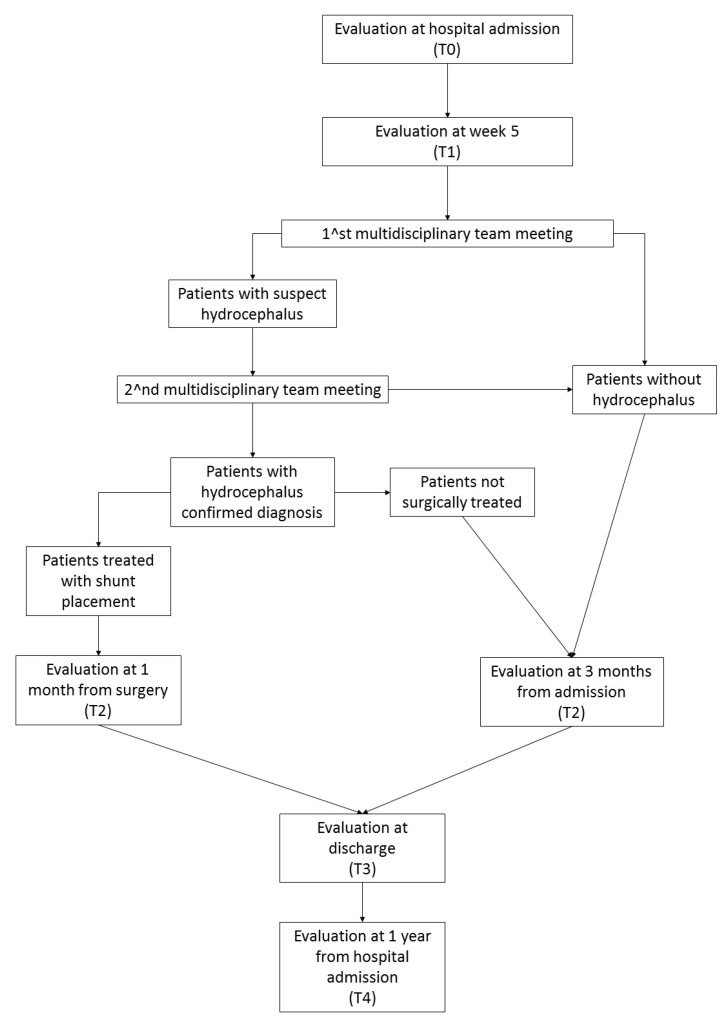
Study flow-chart.

**Table 1 brainsci-13-01067-t001:** Clinical parameters, scales and exams of the study.

Type of Evaluation	
Clinical parameters	- Neurological status - Presence of tracheostomy or mechanical ventilation - Feeding - Dysphagia - Sensory deficits - Cognitive and behavioral deficits - Clinical complications
Clinical measurement scales	- Glasgow Coma Scale (GCS) [21] - Disability Rating Scale (DRS) [22,23,24] - Levels of Cognitive Functioning (LCF) [24,25,26,27] - Coma Recovery Scale Revised (CRS-R) [28,29,30] - MiniMental State Evaluation (MMSE) [31,32] - Montreal Cognitive Assessment (MoCA) [33] - Modified Ashworth Scale (MAS) [34] - Modified Barthel Index (MBI) [35,36,37] - Glasgow Outcome Scale (GOS) [38] - Quality of Life after Brain Injury Overall Scale (QOLIBRI-OS) [39,40]
Functional tests	- Timed “Up-and-Go” test (TUG) [41] - 10 m Walking Test (10 mWT) [42] - Tinetti Performance Oriented Mobility Assessment (POMA) [43]
Neurophysiological exams	- Electroencephalograph EEG - Event-related brain potential (p300/ERB) - Somatosensory-evoked potentials to upper limb (SEPs)
Neuroradiological exams	- Brain CT scan - Brain 3T MRI

## Data Availability

The data that support the findings of this study will be available on request from the corresponding author.

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
