# Peer review of "A New Clinical Protocol for a Timely Diagnosis and Treatment of Hydrocephalus in Patients with Severe Acquired Brain Injury"

_brainsci, 2023, doi:10.3390/brainsci13071067_

Round 1

Reviewer 1 Report (Previous Reviewer 2)

Comments and Suggestions for Authors

The authors addressed my concerns.

Author Response

No revision required.

Reviewer 2 Report (Previous Reviewer 1)

Comments and Suggestions for Authors

Measures of effect size for the statistical tests mentioned should be indicated.

It is worth describing multivariable logistic regression analyzes a bit more in-depth.

It is recommended to describe the limitations of the protocol.

Comments on the Quality of English Language

Minor editing of English language required.

Author Response

Our response to the reviewer are listed below

Measures of effect size for the statistical tests mentioned should be indicated.

R: We have now indicated the measures of effect size that we are planning to use on page 5, lines 197-201.

It is worth describing multivariable logistic regression analyses a bit more in-depth.

R: We apologize for being unclear. We have now expanded the section on page 5, lines 201-204.

It is recommended to describe the limitations of the protocol.

R: Thank you for this suggestion, we have now added two limitations on page 7, line 278-281.

This manuscript is a resubmission of an earlier submission. The following is a list of the peer review reports and author responses from that submission.

Round 1

Reviewer 1 Report

Comments and Suggestions for Authors

The abstract does not reflect the research results obtained.

The results were practically not presented at all.

The statistical tests used are not reflected in the results obtained.

Comments on the Quality of English Language

Minor editing of English language required.

Reviewer 2 Report

Comments and Suggestions for Authors

The manuscript “A new clinical protocol for a timely diagnosis and treatment of hydrocephalus in patients with severe acquired brain injury” by Cava et al is a study protocol article which designed a novel clinical diagnostic protocol for hydrocephalus in patients with severe acquired brain injuries (sABI). This protocol tried to integrate clinical, functional, biochemical and neuroradiological assessments to improve the accuracy of its diagnosis in those patients. The authors found the increase in the proportion of diagnosed cases, the reduced incidence of clinical complications and the increase of the rehabilitative outcomes at discharge etc. Therefore, the authors concluded that this clinical diagnostic protocol will result in a more appropriate assessment and timely treatment of secondary hydrocephalus in patients with sABI. Generally, the subject is of interest and scientifically sound and contains essential contents. This paper is also of importance for providing us the important evidence that the proposed diagnostic protocol would be useful to improve hydrocephalus diagnosis and treatment. The manuscript has been well organized and written. However, I have a concern on the paper.

Although the detailed statistical methods are written, there are no information about the statistical results.